# Integration of Cell-Free DNA End Motifs and Fragment Lengths Can Identify Active Genes in Liquid Biopsies

**DOI:** 10.3390/ijms25021243

**Published:** 2024-01-19

**Authors:** Christoffer Trier Maansson, Louise Skov Thomsen, Peter Meldgaard, Anders Lade Nielsen, Boe Sandahl Sorensen

**Affiliations:** 1Department of Clinical Biochemistry, Aarhus University Hospital, 8200 Aarhus, Denmark; ctm@clin.au.dk (C.T.M.);; 2Department of Clinical Medicine, Aarhus University, 8000 Aarhus, Denmark; 3Department of Biomedicine, Aarhus University, 8000 Aarhus, Denmark; aln@biomed.au.dk; 4Department of Oncology, Aarhus University Hospital, 8200 Aarhus, Denmark; petemeld@rm.dk

**Keywords:** liquid biopsies, fragmentomics, cell-free chromatin immunoprecipitation, gene expression, histone modifications, fragment end motifs

## Abstract

Multiple studies have shown that cell-free DNA (cfDNA) from cancer patients differ in both fragment length and fragment end motif (FEM) from healthy individuals, yet there is a lack of understanding of how the two factors combined are associated with cancer and gene transcription. In this study, we conducted cfDNA fragmentomics evaluations using plasma from lung cancer patients (*n* = 12) and healthy individuals (*n* = 7). A personal gene expression profile was established from plasma using H3K36me3 cell-free chromatin immunoprecipitation sequencing (cfChIP-seq). The genes with the highest expression displayed an enrichment of short cfDNA fragments (median = 19.99%, IQR: 16.94–27.13%, *p* < 0.0001) compared to the genes with low expression. Furthermore, distinct GC-rich FEMs were enriched after cfChIP. Combining the frequency of short cfDNA fragments with the presence of distinct FEMs resulted in an even further enrichment of the most expressed genes (median = 37.85%, IQR: 30.10–39.49%, *p* < 0.0001). An in vitro size selection of <150 bp cfDNA could isolate cfDNA representing active genes and the size-selection enrichment correlated with the cfChIP-seq enrichment (Spearman r range: 0.499–0.882, *p* < 0.0001). This study expands the knowledge regarding cfDNA fragmentomics and sheds new light on how gene activity is associated with both cfDNA fragment lengths and distinct FEMs.

## 1. Introduction

Cell-free DNA (cfDNA) in liquid biopsies has been studied extensively in recent years to improve the sensitivity of methods for detecting somatic variants [1,2,3,4]. Plasma cfDNA originating from tumors is known as circulating tumor DNA (ctDNA), and mutations in ctDNA are important biomarkers for cancer therapies [5,6,7]. However, the use of liquid biopsies is limited by a relatively low ctDNA fraction, below 0.01% in some cases [8]. ctDNA is most frequently identified by somatic mutations, but other features of ctDNA have been studied to increase the sensitivity of methods for detecting ctDNA in early-stage cancers [9,10,11] and minimal residual disease [4,12], for which the ctDNA content is low. These features are related to the fragmentation of cfDNA before circulatory release. First, short cfDNA fragments (<150 bp) contain a larger fraction of mutated fragments than the total cfDNA pool [10,13,14,15,16]. Little is known about why ctDNA is shorter than cfDNA originating from noncancer cells, although the underlying chromatin structure [17,18,19] and the level of the endonuclease, DNASE1L3 [20], are known to affect cfDNA fragmentation. Second, the levels of distinct cfDNA fragment end motifs (FEMs) differ between cancer patients and healthy individuals [9,21,22]. These observations have been used to develop machine learning models to discriminate between plasma samples from healthy individuals and cancer patients [9,21].

In addition, the fragmentation of cfDNA near transcription start sites (TSSs) and transcription-factor binding sites (TFBSs) has been coupled to gene activity [17,18,19,23,24]. With promoter regions in an open chromatin state, active genes are more accessible to endonucleases, leading to lower cfDNA sequencing coverage and a more diverse fragment length profile [23]. Previously, the cell-free chromatin immunoprecipitation (cfChIP) technique was demonstrated to identify active genes in cfDNA. cfChIP offers a new opportunity to identify active genes in liquid biopsies by the immunoprecipitation of trimethylated histone 3 lysine 36 (H3K36me3) nucleosomes from plasma [25,26,27,28].

cfDNA fragmentomics is a growing field in liquid biopsies; however, no comprehensive explanation has been proposed for the relationship between the length of cfDNA or FEMs and gene activity. Next-generation sequencing (NGS) of cfDNA applied to cfChIP has given us an opportunity to create a personal gene expression profile based on liquid biopsies. In the present paper, we took advantage of the ability to obtain information on gene activity and used this information to determine the correlation of gene activity with cfDNA features such as FEM and fragment length (Figure 1). We hereby demonstrated that the combination of cfDNA fragment length and FEM data is superior for predicting the expression patterns of corresponding genes in lung cancer patients and healthy individuals.

## 2. Results

### 2.1. ctDNA Is Shorter Than cfDNA of Noncancer Origin

We performed Cancer Personalized Profiling by deep sequencing (CAPP-seq)cfDNA from twelve stage IV lung cancer patients and seven healthy individuals (Figure 1). The cancer patients had shorter cfDNA than the healthy individuals (Figure 2A). Among the 12 cancer patients, ctDNA was detected in 9/12 patients based on the detection of mutations via CAPP-seq. An analysis of all the cfDNA in mutation-positive cancer patients demonstrated that the length of cfDNA was shorter than that of the cfDNA from mutation-negative patients (Figure 2B). Quantifying the fraction of short cfDNA (<150 bp) revealed similar results for mutation-negative cancer patients and healthy individuals, whereas mutation-positive patients had a greater fraction of short cfDNA fragments than both healthy individuals (*p* = 0.031) and mutation-negative patients (*p* = 0.025) (Figure 2C).

We hypothesize that ctDNA of known cancer cell origin is shorter than cfDNA originating from noncancer cells from the same individual. We combined cfDNA fragments from the nine patients with mutation-positive plasma samples and compared the fragment length of the mutated ctDNA with that of the wild-type (WT) cfDNA fragments at the same genomic position in these samples (Figure 2D, Appendix A). This revealed an increase in mutated fragments with a length less than 150 bp and with a length of approximately 300 bp compared to the corresponding cfDNA. We used in silico size-selection in bins of 10 bp from 50 to 400 bp to evaluate the effect of cfDNA fragment length on the mutant/WT DNA ratio. For each bin, the mutant/WT DNA ratio was normalized to the ratio in the original combined sample (Figure 2E). This revealed the enrichment of mutant ctDNA from 50 to 150 bp and from 200 to 340 bp, whereas WT cfDNA was most prevalent from 150 to 200 bp and from 340 to 400 bp. A pairwise comparison of mutated ctDNA and WT cfDNA also revealed that the abundance of mutated fragments below 150 bp was greater than that of WT cfDNA in the same genomic region (Figure 2F). Collectively, the results in Figure 2 demonstrate that the length of ctDNA with verified mutations is shorter than WT cfDNA from the same samples and that cancer patients have a greater fraction of short cfDNA than healthy individuals.

### 2.2. cfDNA Originating from Genes with High Expression Is Shorter Than cfDNA from Genes with Low Expression

Next, we performed cfChIP-seq on 12 cancer patients. The cfChIP-seq of H3K36me3-containing nucleosomes resulted in the enrichment of cfDNA fragments from genes with active transcription [25,26,27,28,29,30]. A comparison of the cfDNA fragment lengths in cfChIP samples with those in input cfDNA samples from cancer patients revealed the enrichment of shorter cfDNA fragments (Appendix A). Similarly, with a cutoff of 150 bp, we also demonstrated that the fraction under 150 bp was larger in the cfChIP samples than in the matching input samples (Figure 3A, *p* = 0.0093). These results suggest that increased gene activity contributes to the circulatory release of shorter cfDNA fragments. To verify that cfChIP did not result in the artificial trimming of cfDNA fragments, we established two gene sets for each patient representing highly expressed and genes with low levels of expression. We used these gene sets to filter reads from the input cfDNA sample and analyzed the cfDNA fragment lengths of the two gene groups (Appendix A). Quantifying the fraction under 150 bp in the two gene sets revealed that highly expressed genes were systematically represented with shorter cfDNA fragments compared to genes with low expression (Figure 3B, *p* = 8.1 × 10^−6^). Results from healthy individuals (*n* = 4) revealed a similar pattern (Appendix A), implying that the shortening of cfDNA from active genes is not cancer-specific but is related to cfDNA biology in all individuals. cfChIP data could be obtained from four of the seven healthy individuals as the remaining three healthy individuals had insufficient cfChIP yields for an NGS analysis.

Next, we grouped all genes in the CAPP-seq panel into 10 quantiles (Q) based on cfChIP enrichment for each patient (*n* = 12). In each quantile, the fraction of input cfDNA less than 150 bp was estimated and normalized to Q1, which contained the genes with the lowest cfChIP-based activity (Figure 3C). A stepwise increase in gene activity (Q1 to Q10) resulted in a greater fraction of short fragments (Friedman’s ANOVA: *p* < 0.0001). Q10 had a median increase in the <150 bp fraction of 19.99% [IQR: 16.94–27.13%, *p* < 0.0001] compared to that of Q1. These findings were also observed in the validation dataset comprising of healthy individuals (Figure 3D). Here, Q10 had a median increase of 16.01% [IQR: 15.60–16.83%, *p* < 0.0001] in the <150 bp fraction compared to inactive genes. To validate these findings, we performed both in vitro size selection and cfChIP-seq on 11 lung cancer plasma samples (Figure 1). Compared with the input samples, the size selection resulted in a large increase in the <150 bp cfDNA fraction (Appendix A). Following sequencing, the gene enrichment with size selection was estimated using the same gene enrichment applied to the cfChIP samples [28]. The genes were divided into quantiles Q1 to Q10 based on cfChIP enrichment, and the normalized size-selection enrichment in each quantile is displayed in Figure 3E. An increase in gene activity (Q1 to Q10) resulted in greater in vitro size-selection enrichment (Friedman’s ANOVA: *p* < 0.0001). The Q10 median increase in vitro size-selection enrichment was 158.84% (IQR: 125.29–170.11%, *p* < 0.0001) compared to Q1. A correlation of cfChIP gene enrichment with in vitro size-selection gene enrichment revealed a positive correlation (Spearman r range: 0.499–0.882, *p* < 2.2 × 10^−16^; Figure 3F, Appendix A). This finding indicates that the in vitro size selection of plasma cfDNA can be used to predict gene activity from a liquid biopsy. In silico size selection at different fragment length cutoffs also resulted in a positive correlation with cfChIP (Appendix A). The highest correlation was observed for the fraction of fragments less than 160 bp (Spearman r median: 0.403, range: 0.158–0.528), whereas the correlation was reduced for cutoffs above 160 bp.

We hypothesize that the underlying chromatin structure could explain the differences in fragment lengths between active and inactive genes. To investigate this possibility, we estimated the cleavage of cfDNA relative to nucleosome centers which were determined in a previous study [24]. Figure 3G displays the frequency of fragment ends in the core and linker regions of a nucleosome for highly expressed genes and genes with low levels of expression. Compared with the genes with low levels of expression, the highly expressed genes have an increased core/linker end fraction (*p* = 0.0004, Figure 3H). This finding demonstrates that gene activity results in changes in chromatin structure which are reflected in cfDNA fragment end distributions and fragment lengths.

### 2.3. The Active Genes Have Distinct cfDNA FEMs

A comparison of the cfDNA fragment end motifs in the input samples from healthy individuals (*n* = 7) and cancer patients (*n* = 12) revealed differences between the two types of samples (Figure 4A). In addition, we compared the FEM frequencies between paired input and cfChIP samples. These findings demonstrated that 24 FEMs were enriched in the cfChIP samples from cancer patients (Figure 4B), healthy individuals (Appendix A), and all individuals combined (Appendix A). Uniform Manifold Approximation and Projection (UMAP) embedding based on FEM frequencies in input and cfChIP samples from both cancer patients and healthy individuals resulted in distinct clusters (Figure 4C). Samples from healthy individuals were grouped together but were also separated into input and samples containing active genes based on cfChIP enrichment. Moreover, cancer input and cancer cfChIP samples were also separated into two clusters. This finding is in accordance with our results that cfChIP and input samples have distinct FEM frequencies (Figure 4B). The genes associated with low expression and high expression also had a distinct enrichment of FEMs (Figure 4D, Appendix A). The FEMs enriched in the cfChIP samples were generally GC-rich compared to those enriched in input samples (Appendix A). Similarly, compared with genes with low expression, genes with high expression were also enriched in GC-rich FEMs (Appendix A). Interestingly, a significant overlap in enriched FEMs for cfChIP samples and highly expressed genes was observed (*p* = 5.08 × 10^−6^). UMAP embedding FEM frequencies in genes with low and high expression separated the two gene sets into two groups (Figure 4E). The two clusters represent genes with low and high expression, respectively, indicating that cfDNA fragments from genes with low expression have distinct FEMs compared to cfDNA originating from active genes.

### 2.4. Short Fragments Have Distinct FEMs and Originate from Active Genes

To determine whether subsets of the 64 possible 3 bp long FEMs of cfDNA could predict gene expression, we defined the 24 FEMs enriched in the 12 lung cancer cfChIP samples as “cfChIP motifs” (Figure 4B). The nine FEMs enriched in the input samples were defined as “input motifs”. For each lung cancer patient, we grouped all genes based on cfChIP quantiles and quantified the fraction of cfDNA fragments with cfChIP and input motifs (Figure 5A,B). The fraction of fragments with cfChIP motifs increased from Q1 to Q10 (Friedman’s ANOVA: *p* < 0.0001). Q10 had a median increase in the fraction of fragments with cfChIP motifs of 10.87% [IQR: 5.95–15.78%, *p* < 0.001] compared to that of Q1 (Figure 5A). In contrast, the fraction of fragments with input motifs was consistent from Q1 to Q10 (Friedman’s ANOVA: *p* = 0.48; Figure 5B). Similarly, we defined high-expression and low-expression motifs based on the FEMs enriched in highly expressed genes and genes with low levels of expression, respectively (Figure 4D). The fraction of input cfDNA fragments with motifs from highly expressed genes increased from Q1 to Q10 (Friedman’s ANOVA: *p* < 0.0001, Appendix A). Correspondingly, the fraction of fragments with motifs from genes associated with low expression decreased from Q1 to Q10 (Friedman’s ANOVA: *p* < 0.0001; Appendix A). Given that distinct FEMs were identified in active genes and that active genes are enriched in short cfDNA fragments, we investigated the FEMs in size-selected samples compared to unselected samples (Figure 5C). Many of the FEMs enriched in the size-selected samples were GC-rich, similar to the cfChIP motifs, and 12/15 (80%) FEMs enriched in the size-selected samples were also enriched in the cfChIP samples (*p* = 0.00018; Figure 5D). These results indicate that cfDNA originating from active genes contains more short fragments (<150 bp) and a specific subset of FEMs. We combined this knowledge to quantify the fraction of input cfDNA fragments shorter than 150 bp and with cfChIP motifs in gene quantiles based on cfChIP enrichment (Figure 5E). A stepwise increase was observed from Q1 to Q10 (Friedman’s ANOVA: *p* < 0.0001). This indicates that increases in gene activity affect cfDNA fragmentation, resulting in short cfDNA molecules with distinct FEMs. Q10 had a median increase in the fraction of fragments less than 150 bp and with cfChIP motifs of 37.85% [IQR: 30.10–39.49%, *p* < 0.0001] compared to Q1. Combining both fragment lengths and FEMs resulted in greater enrichment in Q10 than fragment lengths alone (median increase = 16.78%, IQR: 8.60–18.20%, *p* < 0.0001; Figure 3C) or cfChIP motifs alone (median increase = 20.35%, IQR: 16.87–32.40%, *p* < 0.0001; Figure 5A). Similar results were observed in the validation dataset comprising healthy individuals (Figure 5F). Here, the median increase in Q10 compared to that in inactive genes was 18.84% [IQR: 17.49–20.07%, *p* < 0.0001], which was also greater than when evaluating fragment lengths alone (median increase = 2.16%, IQR: 1.18–3.51%, *p* < 0.0001; Figure 3D) or FEMs alone (median increase = 15.76%, IQR: 14.40–16.48%, *p* < 0.0001; Appendix A).

## 3. Discussion

In this study, we conducted cfChIP-seq on plasma from non-small cell lung cancer (NSCLC) and small cell lung cancer (SCLC) patients to generate a personal gene expression profile for each individual. In a previous study, we described how cfChIP-seq on these patients can be used to study gene expression differences between the two types of lung cancers [28]. We demonstrated how cfChIP-seq on NSCLC patients resulted in the enrichment of genes known to be upregulated in NSCLC, such as *EGFR* and *RIN3,* whereas SCLC patients had increased gene enrichment of, e.g., *KIF19* and *SMAD4*, which are upregulated in SCLC tumors. This highlights the fact that cfChIP-seq results reflect tumor gene expression and the fragmentomic features identified in this study arise from the tumor cells. 

Previous studies used cfDNA fragment end motif frequencies in machine learning algorithms for early cancer detection and treatment response evaluation [9,21,22,31]. In this study, we focused on the effect of gene activity on FEMs. In a recent study [32], the authors linked FEM frequencies to the methylation status of cfDNA fragments, which indirectly reflects the gene activity in the cell of origin. The authors demonstrated how FEM frequencies vary at different CpG sites, reflecting the methylation status, and they used this knowledge to perform a tissue of origin analyses. It is likely that the FEMs identified in our study that are connected to gene activity are also affected by the methylation of the gene. Future studies are required to expand the body of knowledge concerning the associations between gene activity, methylation status, and cfDNA FEMs.

The ability of cfDNA fragmentomics to infer gene expression has been developed in recent years. These studies have focused primarily on the cfDNA fragmentation profiles surrounding TSS [19,23], open chromatin regions [17], and TFBSs [18,33]. Given that the H3K36me3 modification, which was used in our cfChIP assays, is located primarily in the gene body of active genes [29,30], and that the applied NGS panel is not focused around the TSS [28], our results preferentially reflected the cfDNA fragmentation within the gene body. Nevertheless, we observed that increased gene activity results in shorter cfDNA fragments, which is in accordance with the findings of a previous study [23]. Our results, based on cfChIP-seq, were verified using an in vitro size-selection enrichment of cfDNA, as the results from these two approaches were found to be correlated. Combining FEMs with fragment lengths increased the association with cfDNA fragments originating from active genes, indicating that increased gene activity results in both shorter fragments and distinct fragment end motifs. FEMs are a new field in liquid biopsy research, and multiple factors, including nuclease activity [20,34,35] and methylation [32], can affect FEMs. Future whole genome sequencing (WGS)-based studies of FEMs and cfChIP in combination with in vitro cfDNA size selection could further validate our findings and provide comprehensive insights into the relationship between cfDNA fragmentomics and underlying gene activity.

cfDNA fragment size distribution differs between cancer patients and healthy individuals [4,15,36], which was also observed in this study. Previous studies have identified how cancer patients have increased amounts of short fragments [4,15] and high molecular fragments caused by cell necrosis [37]. In addition, we evaluated the lengths of the mutated ctDNA fragments and compared them to WT cfDNA fragments in the same patient. A limitation of these analyses is that only 9 of the 12 lung cancer patients were mutation-positive and the number of ctDNA mutations was low (median = 3, range 0–6). However, we are able to demonstrate an enrichment of mutated ctDNA in short cfDNA fragments. This result is in alignment with a previous study [10].

A limitation of our study is the relatively few individuals included (twelve cancer patients and seven healthy individuals) and the size of our gene panel used for targeted NGS. In addition, only stage IV lung cancer patients with a high ctDNA fraction were analyzed. As a result, our findings need to be repeated in larger patient cohorts and expanded to include other types of cancer, patients of different ethnicities and ages, as well as the full spectrum of cancer stages. Likewise, it would be interesting to perform these experiments using a broader NGS approach, such as WGS, to generalize the information obtained from cfDNA fragmentomics. Despite this, we were able to replicate our findings in cfDNA from healthy individuals using WGS data from a previous study [24] in which the expected gene activity was based on data from peripheral blood mononuclear cells (PBMCs). These findings suggest that our results are applicable to other types of NGS methods and that the fragmentomic features identified in our study can be generalized to other samples. Future studies will demonstrate how gene expression based on liquid biopsies can be used to monitor tumor development and identify treatment responses using longitudinal plasma samples from the same individuals.

## 4. Materials and Methods

### 4.1. Plasma Samples

Plasma samples from twelve stage IV lung cancer patients diagnosed with either NSCLC (*n* = 8) or SCLC (*n* = 4) and from seven healthy individuals were used. Peripheral blood was drawn in 10 mL EDTA tubes from each individual and centrifuged at room temperature within 2 h at 1400× *g* for 15 min. Plasma was aliquoted and stored at −80 °C. The plasma was split and subjected to (1) input cfDNA purification, (2) cfChIP, or (3) cfDNA purification and in vitro size selection (Figure 1). One NSCLC patient had insufficient remaining plasma for in vitro size selection, and therefore, 11/12 lung cancer patients were subjected to in vitro size selection. All three types of samples were subjected to NGS via CAPP-seq [2], as described in detail below. The amount of plasma and the resulting number of reads for each sample are presented in Appendix A.

### 4.2. Cell-Free Chromatin Immunoprecipitaiton (cfChIP) Enrichment

cfChIP-seq was performed as described by Maansson et al. [28] (Figure 1, left). H3K36me3 cfChIP enrichment was calculated for each gene, and the results were used as a surrogate for gene activity [25,27]. In brief, the plasma was cleared of circulating antibodies using empty protein A/G magnetic beads (ThermoFisher Scientific, 88802, Waltham, MA, USA). The cleared plasma was added to protein A/G magnetic beads coated with anti-H3K36me3 antibodies (Abcam 9050, Cambridge, United Kingdom) and incubated overnight at 4 °C. The beads were then washed, and the precipitated DNA was eluted. The cfChIP sample and the input plasma cfDNA were purified using an Apostle MiniMax High Efficiency cfDNA Isolation Kit (Beckman Coulter, Indianapolis, IN, USA) and subjected to CAPP-seq. Deduplicated cfChIP BAM files were used for a H3K36me3 enrichment analysis. For each gene, the number of reads was normalized to the total number of reads and the NGS coverage of that particular gene [28].

### 4.3. In Vitro Size Selection

A PippinHT (Sage Science, Beverly, MA, USA) instrument was used to perform the in vitro size-selection of cfDNA (Figure 1, right). First, 20 µL of purified cfDNA from each plasma sample was added to a 3% agarose gel cassette. The range mode of the system was set to collect cfDNA within 95–230 bp with a pause at 152 bp, resulting in the collection of two cfDNA fractions. The two fractions corresponded to cfDNA fragments with lengths of 95–152 bp and 152–230 bp, respectively. The cfDNA fragments were eluted in 30 µL of electrophoresis buffer. The fraction with short fragments (95–152 bp) was subjected to CAPP-seq.

### 4.4. CAPP-Seq

CAPP-seq is based on hybridization-capture NGS [2]. The sequencing library was prepared using the AVENIO ctDNA surveillance kit (Roche Sequencing Solutions, Mannheim, Germany), covering 198 kb in 197 hypermutated genes [38]. The gene fragments were sequenced using a NextSeq 500 (Illumina, San Diego, CA, USA), and the data were analyzed using the AVENIO Oncology Analysis Software (v. 2.0.0). Deduplicated BAM files based on unique molecular identifiers (UMIs) were used for a cfChIP enrichment analysis, whereas position-deduplicated BAM files were used to estimate fragment lengths and FEMs.

### 4.5. ctDNA Detection

Mutations in cfDNA were identified using the AVENIO Oncology Analysis software v. 2.0.0 (Roche Sequencing Solutions, Mannheim, Germany), as described previously [39]. The excluded variants included synonymous variants and variants present in the Exome Aggregation Consortium (ExaC) with an allele frequency > 0.10%, the 1000 Genomes Project, or the Single Nucleotide Polymorphism database (dbSNP Common). The included variants needed to be present in the Catalog of Somatic Mutations in Cancers (COSMIC) database, The Cancer Genome Atlas (TCGA), or the Loci of Interest list. Mutated cfDNA fragments were isolated as described previously [12], and fragments covering the same genomic position without the mutation were defined as WT fragments.

### 4.6. Fragment Length and FEM Analyses

Position-deduplicated BAM files were used for fragment length and FEM analyses. Fragment lengths were estimated using the start and end positions of paired-end sequencing reads. Reads with no fragment length information or a fragment length of 0 were excluded from the analysis. Reads containing N nucleotides in either of the fragment termini were removed. The frequencies of 5′-3′ 3-mers at each end of a fragment were determined. The frequencies at both ends of the fragments were combined, and the fraction of each possible 3-mer (*n* = 4^3^ = 64) for each sample was calculated. The FEM frequencies for all 64 possible motifs were used for a Uniform Manifold Approximation and Projection (UMAP) analysis.

### 4.7. High/Low-Expressed Genes and cfChIP Quantiles

Gene expression was estimated using cfChIP-seq for each individual, and the 15 most enriched genes were considered to be highly expressed, whereas the 15 least enriched genes were considered to be expressed at low levels. Furthermore, genes were grouped in cfChIP quantiles for each patient. Each quantile contained 19 or 20 genes such that Q1 contained the genes with the lowest enrichment and Q10 contained the most enriched genes. Reads from cfDNA input samples in either high/low-expressed genes or cfChIP quantiles were filtered, and fragment lengths and FEMs were analyzed.

### 4.8. Nucleosomal Positioning Analysis

For each cfDNA fragment in high- or low-expressed genes, we estimated the distance from the nearest nucleosome center to the cfDNA fragment ends as described previously [15]. Information regarding nucleosome positions was based on the CH01 nucleosome track from Snyder et al. [24]. This track is based on the pooled whole-genome sequencing of cfDNA from multiple healthy individuals. We identified input cfDNA fragments overlapping with nucleosome centers and calculated the distance from each nucleosome center to the start and end of the fragment. For fragments overlapping with multiple nucleosome centers, only the nearest nucleosome to each start and end was considered. We defined the nucleosome core as −75 to +75 bp relative to the nucleosome dyad and the linker region as −75 to −95 or +75 or +95 bp relative to the nucleosome dyad. Based on these regions, we calculated a core/linker end fraction as the number of fragment ends in the core region relative to the number of fragment ends in the linker region.

### 4.9. Validation Data

cfDNA WGS data published by Snyder et al. [24] were used to validate the findings in this study. The BH01 (GSM1833219) cfDNA-seq dataset representing the deep WGS of an unknown number of healthy individuals was down sampled to 20 0.9X WGS samples. This dataset is referred to as the “validation dataset”. Furthermore, RNA-seq data from PBMCs (GSE107011, *n* = 13) [40] were used to determine gene expression patterns in healthy individuals. Reads in the validation dataset were grouped into gene sets based on PBMC gene activity. Genes with no expression (log_2_(TPM + 1) = 0 in all samples) were classified as “inactive” (*n* = 2262). The remaining genes were grouped into quantiles with 1747 to 1751 genes in each quantile. Q1 contained the genes with the lowest average log_2_(TPM + 1), and Q10 contained the genes with the highest average log_2_(TPM + 1).

### 4.10. Statistical Analysis

All the data analyses were performed in R version 4.3.0. BAM files were loaded into R using the chromstaR [41] package. cfDNA fragments were analyzed using the GenomicRanges [42] Biostrings [43], and Rsamtools [44] packages. Plots and graphs were created using the dplyr [45], ggpubr [46], and ggplot2 [47] packages. Differences in both fragment lengths and FEMs between the input and cfChIP samples were evaluated with a paired *t*-test. A comparison of highly expressed genes and genes with low levels of expression from the same individual was also performed with a paired *t*-test. Differences between cancer patients and healthy individuals as well as between mutation-positive and mutation-negative patients were tested using an unpaired *t*-test. Global differences in fragmentation for cfChIP quantiles were tested using Friedman’s ANOVA, and if *p* < 0.05, Q2–Q10 were compared individually to Q1 using paired *t*-tests. For the FEM analysis, *p*-values were adjusted for multiple testing using the false discovery rate (FDR) [48], and *q*-values < 0.05 were considered significant. Significant overlap between FEM sets was determined using a hypergeometric test, where *p* < 0.05 was considered to indicate statistical significance.

## 5. Conclusions

In this study, we identified key fragmentomic features related to cfDNA from active genes. Based on the cfChIP-seq of cfDNA, which is a novel technique for measuring gene activity in liquid biopsies, we created a personal gene expression profile and used this information to detect differences in fragmentomics between highly expressed genes and genes with low levels of expression. Here, we demonstrated that active transcription is associated with shorter cfDNA fragments and more GC-rich FEMs. Interestingly, combining the frequency of short fragments and distinct FEMs resulted in greater enrichment in highly expressed genes than evaluating fragment lengths or FEMs separately. This was also observed in the validation dataset from healthy individuals. This study provides novel insights into the biology behind cfDNA fragmentomics and demonstrates that FEMs are an important new characteristic of cfDNA which can increase the utility of liquid biopsies for determining gene activity under various pathological conditions.

## Figures and Tables

**Figure 1 ijms-25-01243-f001:**
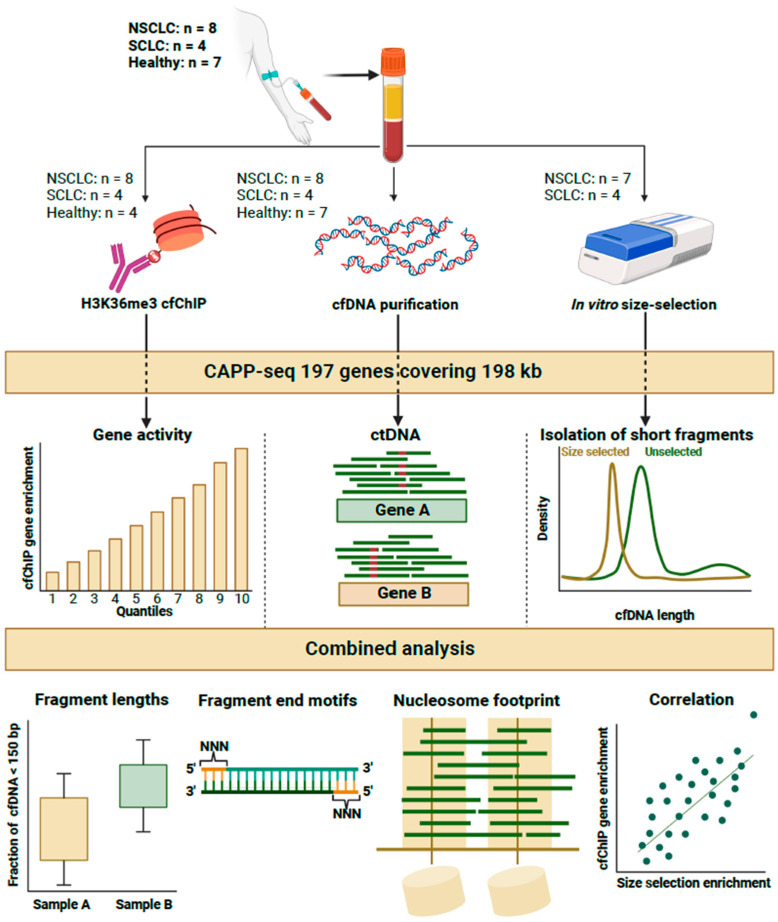
Experimental setup of the study. Plasma samples from 12 lung cancer patients and 7 healthy individuals were subjected to input cfDNA purification (**middle**), H3K36me3 cfChIP (**left**), or in vitro size selection (**right**). Thes samples were sequenced via Cancer Personalized Profiling by deep sequencing (CAPP-seq) 197 genes. cfChIP samples were used to estimate gene activity for each individual. The input cfDNA samples were used to detect mutations in cell-free DNA. In vitro size selection was used to investigate the properties of short cfDNA fragments. The data from each type of sample were analyzed in relation to fragment length, fragment end motifs (FEMs), nucleosomal footprint, and enrichment correlation.

**Figure 2 ijms-25-01243-f002:**
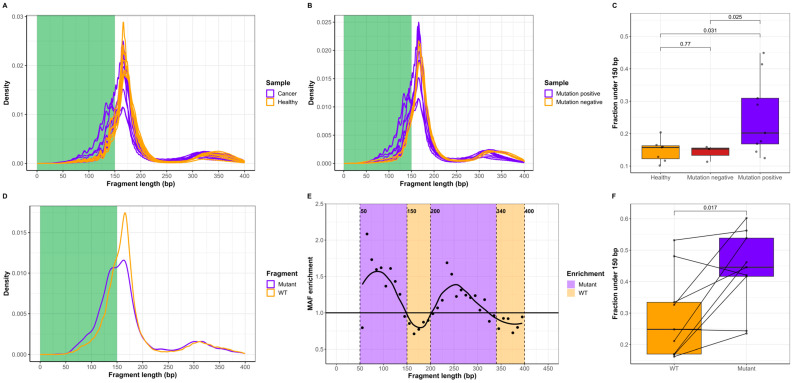
cfDNA fragment lengths in cancer patients and healthy individuals. (**A**) The cfDNA fragment size distribution in lung cancer patients (*n* = 12, purple) and healthy individuals (*n* = 7, orange). Short cfDNA (0–150 bp) is marked in green. (**B**) cfDNA fragment size distribution in mutation-positive (*n* = 9, purple) and mutation-negative (*n* = 3, orange) lung cancer patients. (**C**) Fraction of cfDNA shorter than 150 bp in healthy individuals (*n* = 7, orange), mutation-negative (*n* = 3, red) and mutation-positive (*n* = 9, purple) lung cancer patients. Groups were compared using an unpaired *t*-test. (**D**) Size distribution of mutated and WT cfDNA fragments in the 9 patient samples containing ctDNA mutations. (**E**) The relative mutant/WT cfDNA fraction in bins (10 bp) from 50 to 400 bp. Each point represents the relative mutant/WT cfDNA enrichment compared to the molecular allele fraction (MAF) of the unselected cfDNA pool. Purple indicates cfDNA fragment lengths with mutant enrichment, and orange represents WT cfDNA enrichment. (**F**) Pairwise comparison of the fractions under 150 bp for WT cfDNA (*n* = 9, orange) and mutated fragments (*n* = 9, purple). Groups were compared using a paired *t*-test.

**Figure 3 ijms-25-01243-f003:**
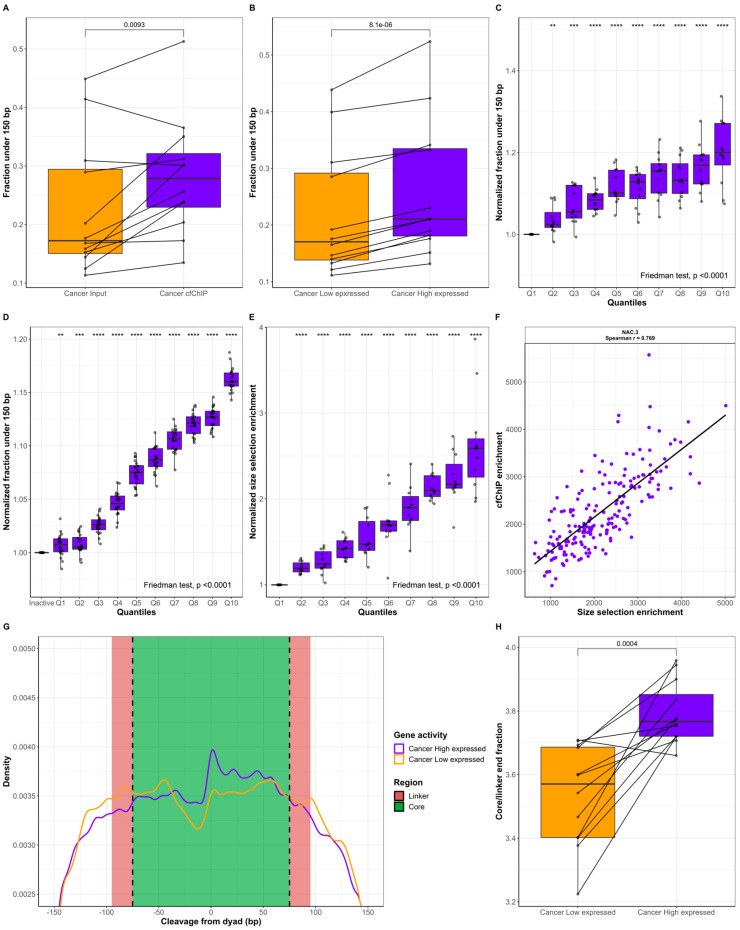
Fragment length analysis of cfDNA in active genes. (**A**) Pairwise comparison of the fraction under 150 bp for input (*n* = 12, orange) and cfChIP (*n* = 12, purple) samples. (**B**) Pairwise comparison of the fraction under 150 bp for genes with low expression (*n* = 12, orange) and high expression (*n* = 12, purple). (**C**) Normalized fraction under 150 bp for genes in the cfChIP quantiles Q1 to Q10 for lung cancer patients (*n* = 12). (**D**) Normalized fraction under 150 bp for cfDNA in healthy individuals. The fragments are grouped as inactive or in quantiles Q1 to Q10. (**E**) Normalized in vitro size-selection enrichment for genes in the cfChIP quantiles Q1 to Q10 for lung cancer patients (*n* = 11). For (**C**,**E**), each quantile is normalized to the fraction in Q1, whereas quantiles in (**D**) are normalized to inactive genes. (**F**) Representative example of cfChIP enrichment compared to size-selection enrichment. NAC.3 represents the median Spearman’s correlation between the 11 samples. (**G**) Distribution of fragment end cleavage relative to the nucleosome dyad in lung cancer patients (*n* = 12). Purple represents highly expressed genes and orange represents genes with low levels of expression. (**H**) Pairwise comparison of the core/linker end fraction for genes with low expression (*n* = 12, orange) and high expression (*n* = 12, purple). For (**A**,**B**,**H**), groups were compared using a paired *t*-test. **: *p* ≤ 0.01, ***: *p* ≤ 0.001, ****: *p* ≤ 0.0001.

**Figure 4 ijms-25-01243-f004:**
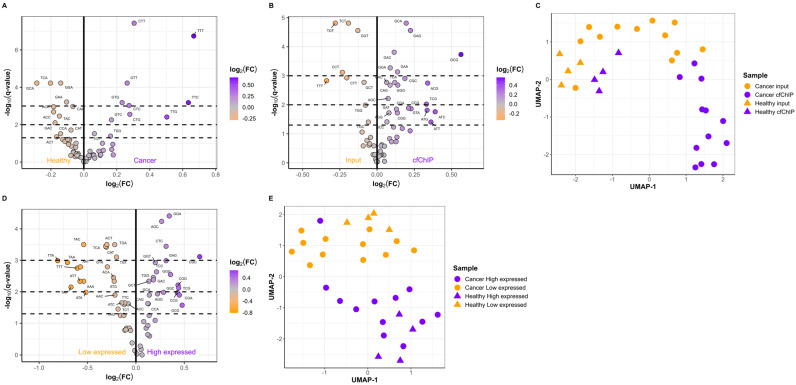
Active and inactive genes can be distinguished based on fragment end motifs (FEMs). (**A**) Differences in FEM frequencies between cancer patients (*n* = 12) and healthy individuals (*n* = 7). (**B**) Differences in FEMs in paired input and cfChIP cancer samples (*n* = 12). (**C**) Uniform Manifold Approximation and Projection (UMAP) based on FEM frequencies in the input and cfChIP samples. (**D**) Differences in FEMs for genes expressed at low and high levels in cancer input samples (*n* = 12). (**E**) UMAP based on the FEM frequencies in genes expressed at low and high levels. For (**A**,**B**), and (**D**), FEMs with a *q*-value less than 0.05 are labeled with the motif. Dashed lines indicate *q* = 0.05, *q* = 0.01, and *q* = 0.001. For (**C**,**E**), cancer patients are represented as circles, and healthy individuals are represented as triangles. The colors indicate the type of sample.

**Figure 5 ijms-25-01243-f005:**
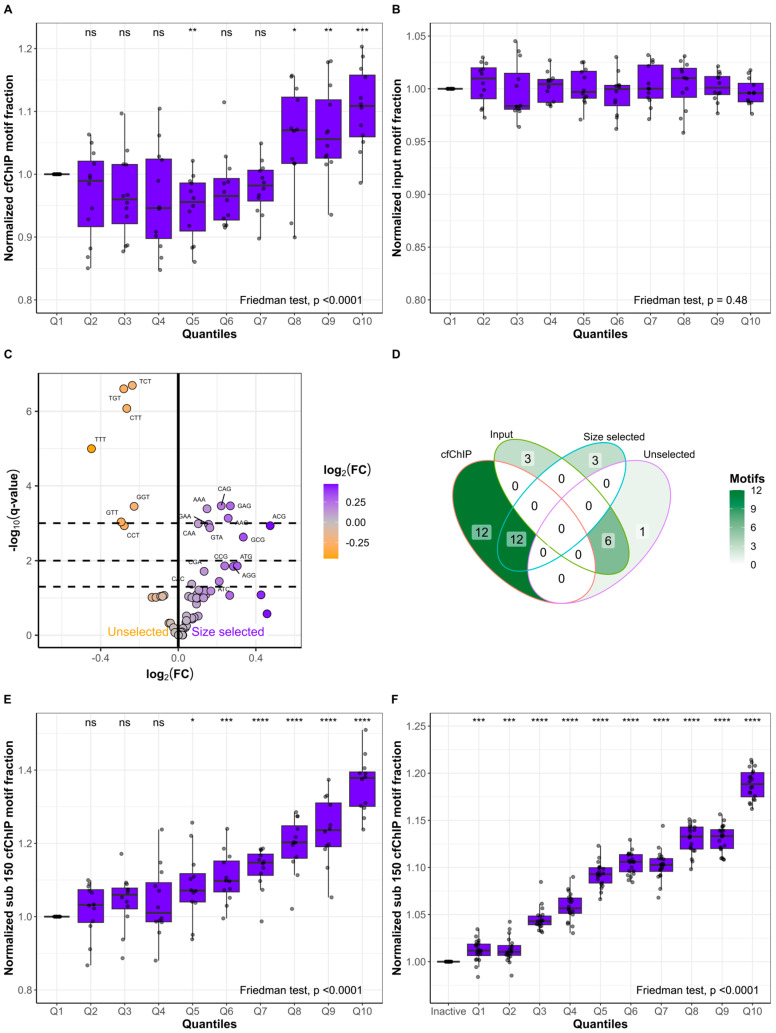
Fragment end motifs from short fragments predict gene expression. For each lung cancer patient (*n* = 12), the genes in the cfChIP quantiles were determined. For each quantile, the fraction of fragments with fragment end motifs enriched in cfChIP (**A**) or input (**B**) samples was calculated. (**C**) Differences in fragment end motifs in paired unselected and size-selected cancer samples (*n* = 11). (**D**) Venn diagram representing the overlap in significant fragment end motifs in the analyses of cfChIP compared to input samples and unselected compared to size-selected samples. (**E**) Normalized fraction of fragments less than 150 bp with cfChIP motifs for genes in cfChIP quantiles Q1 to Q10 for lung cancer patients (*n* = 12). (**F**) Normalized fraction of fragments less than 150 bp with cfChIP motifs for genes in the validation dataset. The fragments are grouped as inactive or in quantiles Q1 to Q10. For (**A**,**B**,**E**), each quantile is normalized to the fraction in Q1, whereas quantiles in (**F**) are normalized to inactive genes. ns: *p* > 0.05, *: *p* ≤ 0.05, **: *p* ≤ 0.01, ***: *p* ≤ 0.001, ****: *p* ≤ 0.0001.

## Data Availability

This study used the following publicly available datasets: RNA-seq of PBMCs (GSE107011), WGS of cfDNA in healthy individuals (GSM1833219; BH01), and nucleosome position information based on the cfDNA WGS of healthy individuals (GSE71378; CH01). The cfChIP-seq data are from a previous study from our group [28]. The remaining data associated with this study are available upon request from the corresponding author. The data are not publicly available due to privacy concerns.

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
