# Peer review of "Integration of Cell-Free DNA End Motifs and Fragment Lengths Can Identify Active Genes in Liquid Biopsies"

_ijms, 2024, doi:10.3390/ijms25021243_

Round 1

Reviewer 1 Report

Comments and Suggestions for Authors

The paper presents an approach to analyzing cell-free DNA (cfDNA) in plasma from lung cancer patients and healthy individuals. By integrating the analysis of cfDNA fragment lengths and fragment end motifs (FEMs), the study demonstrates that highly expressed genes are associated with shorter cfDNA fragments and distinct GC-rich FEMs. Utilizing cell-free chromatin immunoprecipitation sequencing (cfChIP-seq), the research provides insights into cfDNA fragmentomics, suggesting its potential utility in cancer diagnosis and treatment monitoring. Here are several comments, observations on limitations, and suggestions for improvement:

  1. The resolution of all figures in the manuscript is too low, especially Figure 3 and 5, making it very hard to read the legend and axis-label
  2. In Figure 3, panels E and F have a sample size (n = 11) that appears inconsistent with other data presentations. An explanation for this discrepancy would aid in understanding the data better.
  3. The manuscript lacks a dedicated "Conclusion" section. 
  4. The study's small sample size and lack of population diversity limit the generalizability of the findings. A larger and more varied cohort to is needed to enhance the robustness and applicability of the results.
  5. The cross-sectional nature of the study provides a snapshot of cfDNA characteristics. However, longitudinal studies that track cfDNA changes over time in the same individuals, particularly in the context of cancer progression or treatment response, would offer invaluable insights. Such an approach would allow for a more dynamic understanding of cfDNA variations and their clinical significance.

Reviewer 2 Report

Comments and Suggestions for Authors

Despite the low sample size, representing the only study limitation, Maansson et al. proposed a deep well-performed liquid biopsy analysis in lung cancer subjects. Actually, results go beyond the specific disease, providing insights applicable to the whole oncology field.  

Authors demonstrated that cfDNA is shorter and has different FEM in oncology subjects as compared with healthy, and that this cfDNA features reflect the expression activity.

I have no major concerns to express about the methodology and the “next-step experiment” approaches, which I consider as excellent.

Minor issues:

-        It is already known that cfDNA derived from the tumor is shorter, as compared with the physiological. Actually, the best interpretation is that the tumor cfDNA is formed through a necrosis process, whereas the physiological cfDNA derive from apoptosis of blood circulating cells. Please discuss about that and cite pertinent literature (e.g. Marotta V, Cennamo M, La Civita E, Vitale M, Terracciano D. Cell-Free DNA Analysis within the Challenges of Thyroid Cancer Management. Cancers (Basel). 2022 Oct 31;14(21):5370. doi: 10.3390/cancers14215370. PMID: 36358788; PMCID: PMC9654679.).

-        I would not define ctDNA+ subjects, but rather subjects with mutation+ liquid biopsy.

Comments on the Quality of English Language

Minor editing. 

Round 2

Reviewer 1 Report

Comments and Suggestions for Authors

The authors have addressed the raised questions appropriately, this manuscript is suitable for publication. 

Author Response

Thank you